# Multi-Omics Analysis of the Effects of Soil Amendment on Rapeseed (*Brassica napus* L.) Photosynthesis under Drip Irrigation with Brackish Water

**DOI:** 10.3390/ijms25052521

**Published:** 2024-02-21

**Authors:** Ziwei Li, Hua Fan, Le Yang, Shuai Wang, Dashuang Hong, Wenli Cui, Tong Wang, Chunying Wei, Yan Sun, Kaiyong Wang, Yantao Liu

**Affiliations:** 1Agricultural College, Shihezi University, Shihezi 832000, China; liziwei09@stu.shzu.edu.cn (Z.L.); fanhua@shzu.edu.cn (H.F.); yl_agr@shzu.edu.cn (L.Y.); wangshuai@stu.shzu.edu.cn (S.W.); hongdashaugn@stu.shzu.edu.cn (D.H.); cuiwenli@xjshzu.com (W.C.); wt@stu.shzu.edu.cn (T.W.); 20201012057@stu.shzu.edu.cn (C.W.); sunvan72@xishzu.com (Y.S.); 2Institute of Crop Research, Xinjiang Academy of Agricultural Reclamation Sciences, Shihezi 832000, China; ziheng1979@126.com

**Keywords:** brackish water, metabolic pathways, energy metabolism, abiotic stresses, multi-omics

## Abstract

Drip irrigation with brackish water increases the risk of soil salinization while alleviating water shortage in arid areas. In order to alleviate soil salinity stress on crops, polymer soil amendments are increasingly used. But the regulation mechanism of a polymer soil amendment composed of polyacrylamide polyvinyl alcohol, and manganese sulfate (PPM) on rapeseed photosynthesis under drip irrigation with different types of brackish water is still unclear. In this field study, PPM was applied to study the responses of the rapeseed (*Brassica napus* L.) phenotype, photosynthetic physiology, transcriptomics, and metabolomics at the peak flowering stage under drip irrigation with water containing 6 g·L^−1^ NaCl (S) and Na_2_CO_3_ (A). The results showed that the inhibitory effect of the A treatment on rapeseed photosynthesis was greater than that of the S treatment, which was reflected in the higher Na^+^ content (73.30%) and lower photosynthetic-fluorescence parameters (6.30–61.54%) and antioxidant enzyme activity (53.13–77.10%) of the A-treated plants. The application of PPM increased the biomass (63.03–75.91%), photosynthetic parameters (10.55–34.06%), chlorophyll fluorescence parameters (33.83–62.52%), leaf pigment content (10.30–187.73%), and antioxidant enzyme activity (28.37–198.57%) under S and A treatments. However, the difference is that under the S treatment, PPM regulated the sulfur metabolism, carbon fixation and carbon metabolism pathways in rapeseed leaves. And it also regulated the photosynthesis-, oxidative phosphorylation-, and TCA cycle-related metabolic pathways in rapeseed leaves under A treatment. This study will provide new insights for the application of polymer materials to tackle the salinity stress on crops caused by drip irrigation with brackish water, and solve the difficulty in brackish water utilization.

## 1. Introduction

Freshwater is scarce in arid regions, so drip irrigation with brackish water is an important measure to alleviate the contradiction between agricultural production and water scarcity [1]. At present, countries around the world mainly use brackish and fresh water mixed irrigation, brackish and fresh water rotation irrigation, and filtration technology to increase the use of brackish water [2]. However, these methods still do not meet freshwater needs due to freshwater shortages, great domestic and industrial freshwater needs, and high costs in most arid regions, and they also lead to the surface accumulation of a large amount of salt, which increases the risk of salinization. Particularly, irrigation with brackish water could significantly alter many physiological processes of plants, especially photosynthesis, resulting in retarded plant growth and great yield reduction [3,4].

The possible ways of salt stress affecting plant photosynthesis mainly include: ion poisoning, osmotic stress, and sugar accumulation-induced feedback inhibition [5]. Previous studies have shown that high soil salinity could destroy the stomatal structure and chloroplast structural integrity of plants [6], reduce net photosynthesis, the photosynthetic pigment content, the F_v_/F_m_ (maximum photochemical efficiency of photosystem II) ratio, stomatal conductance, the transformation rate, gas exchange, the PSII (photosystem II) efficiency, and water potential, and cause osmotic stress and nutrient deficiency in plants [7,8]. Salt stress can also inhibit the carbon assimilation of plants, resulting in a reduced accumulation of photosynthates [9]. In addition, salt stress also impacts the activities of key enzymes related to photosynthesis, such as RuBPCase (ribulose-1, 5-bisphophatecarboxylase), leading to reduced photosynthetic carbon assimilation and photosynthetic rate [10].

At present, there are many methods used to improve the tolerance of crops to salt stress. Biochar, a commonly used solid soil amendment, can reduce the negative effects of drought and salt stress on plants, as well as improve the soil’s physical structure, water holding capacity, and fertility and plant photosynthesis [11]. However, biochar is insoluble in water and cannot be used through a drip irrigation system that is widely applied by farmers in arid areas [12]. The artificial and mechanical application of biochar are time-consuming and costly. Other soil amendments widely studied by scholars such as zeolite, bentonite and gypsum are also insoluble in water [13]. Bioactive compounds and organic amendments such as plant growth-promoting bacteria and biofertilizers can enhance crop salt tolerance and yield by maintaining ionic homeostasis and reducing oxidative damage, but such amendments are expensive and have a slow effect and low utilization rate. Especially, a high pH environment always causes inactivation of most bacteria [14]. Therefore, bioactive compounds and organic amendments are also not suitable for large-scale application in arid areas. Therefore, it is urgent to find a soil amendment suitable for a drip irrigation system. 

PPM is an independently developed soil amendment with inorganic and organic polymers as the main components. It has high-water solubility, and can be applied through drip irrigation [15]. Studies have shown that polyacrylamide has no significant negative effects on aquatic ecosystems and soil organisms when used at recommended doses (10–20 kg·hm^−2^) [16,17]. The biodegradability of PVA depends on its molecular weight and other structural properties [18]. In terms of ecological safety, polyvinyl alcohol is safer, but with increasing decomposition time, there is a need to be concerned about its long-term accumulation in soil and water bodies. The proper use of manganese sulfate avoids manganese deficiency and thus enhances plant growth and resistance, and generally the safe concentration of manganese sulphate solution is 0.1–1% [19,20]. Our previous study found that PPM had a variety of functional groups, such as carboxyl, alcohol hydroxyl, and silicone, whose main effect is to improve the soil structure and buffer saline ions in the soil [15,21]. The application of PPM could improve the resistance of cotton, wheat, and other crops to salt stress [15]. *B. napus* is a salt-tolerant plant. In contrast to salt-tolerant crops such as cotton and wheat, which resist salt damage by “salt avoidance” (avoiding uptake of salt), variety Huayouza 82 is able to take up a large amount of sodium ions and store them in the stalks and petioles [13,22]. It was found that variety Huayouza 82 was able to grow in a salt concentration of 0.8–1.2%; variety Huayouza showed the best salinity tolerance under salt stress at 0.5–1.5% NaCl concentration compared to varieties such as Zheshuang 72, Zheyou 50, Zheda 622, and Feeding Oil No.2 [23]. The cropping of *B. napus* combined with drip irrigation with brackish water can improve the utilization of brackish water, alleviate the freshwater crisis, and ensure agriculture production in arid areas. However, previous study showed that when soil salinity reached 5 g·L^−1^, the growth of *B. napus* was significantly inhibited [24].

Therefore, in this study, the molecular regulatory mechanism of PPM on the photosynthesis of B. napus under drip irrigation with neutral (NaCl) and alkaline (Na_2_CO_3_) water was determined by the integrated transcriptomic and metabolomic analysis. The objectives were to reveal (1) the differences in the effects of drip irrigation with neutral and alkaline water on the physiological and photosynthetic parameters of *B. napus*, and (2) the differences in the mechanisms of PPM improving the photosynthesis of *B. napus* under drip irrigation with neutral and alkaline water. This study will have guiding significance for brackish water drip irrigation and crop yield increase in arid areas.

## 2. Results

### 2.1. Effects of Application of PPM on Biomass and Leaf Physiological Parameters of B. napus

Comparing the ACK (alkaline brackish water irrigation control treatment) group with the SCK (neutral brackish water irrigation control treatment) group, the yield, root, stem, and leaf fresh weight decreased by 13.24%, 17.88%, 9.33%, and 6.01%, respectively (*p* < 0.05). A decrease of 17.96% was observed in the leaf K^+^ content, while there was a rise of 51.32% in the K^+^/Na^+^ ratio, and a decrease of 40.52% in the REC (relative electrical conductivity) in the ACK group (*p* < 0.05), the Na^+^ content increased by 73.30% (*p* < 0.05), compared with those in the SCK group. The application of PPM obviously increased the fresh weight, leaf K^+^/Na^+^ ratio, K^+^ content, and REC content of *B. napus* under brackish water irrigation. The yield, root, stem, and leaf fresh weight in the SPPM (neutral brackish water irrigation and PPM treatment) group increased by 30.19%, 64.38%, 61.60% and 78.98%, respectively (*p* < 0.05) compared with those in the SCK group. The leaf K^+^/Na^+^ ratio and K^+^ content in the SPPM group increased by 29.54% and 12.22%, respectively (*p* < 0.05), while the leaf Na^+^ content and REC decreased by 12.22% and 64.88%, respectively (*p* < 0.05), compared with those in the SCK group. The yield, root, stem, and leaf fresh weight in the APPM (alkaline brackish water irrigation and PPM treatment) group increased by 38.83%, 71.04%, 68.58%, and 73.86%, respectively (*p* < 0.05) compared with those in the ACK group, but it decreased by 22.68%, 13.67%, and 5.91%, respectively (*p* < 0.05) compared with those in the SPPM group. The leaf K^+^/Na^+^ ratio and K^+^ content in the APPM group increased by 36.14% and 26.55%, respectively (*p* < 0.05), while the leaf Na^+^ content and REC decreased by 7.05% and 22.23%, respectively (*p* < 0.05), compared with those in the ACK group (Figure 1).

### 2.2. Effects of PPM Application on Photosynthetic Parameters of B. napus Leaves

The Pn (leaf net photosynthetic rate), Gs (stomatal conductance), and Tr (transpiration rate) of *B. napus* leaves in the SPPM group increased by 10.55%, 17.01%, and 34.06%, respectively (*p* < 0.05), but the C_i_ (intracellular CO_2_ concentration) decreased by 3.36% (*p* > 0.05), compared with those in the SCK group. The Pn, Gs, and Tr in the APPM group increased by 17.01%, 23.74%, and 42.34%, respectively (*p* < 0.05), but the C_i_ decreased by 18.83% (*p* < 0.05), compared with those in the ACK group. In addition, there are no differences in Pn, Tr, and Gs between the S (SCK vs. SPPM treatment) groups and A (ACK vs. APPM treatment) groups (*p* > 0.05) (Figure 2a). The leaf Chl a (chlorophyll a) in the ACK group increased by 10.00% (*p* < 0.05), but the Chl b (chlorophyll b) and Car (carotenoids) reduced by 14.49% and 16.74%, respectively, compared with those in the SCK group. The application of PPM enhanced photosynthetic pigment contents of *B. napus* leaves under drip irrigation using brackish water. The leaf Chl a, Chl b, Chl a + Chl b, and Car in the SPPM group increased by 56.34%, 40.05%, 51.68%, and 33.83%, respectively (*p* < 0.05) compared with those in the SCK group, and those in the APPM group increased by 35.00%, 43.25%, 36.97%, and 62.52%, respectively (*p* < 0.05) compared with those in the ACK group. In addition, the Chl a in the SPPM group decreased by 6.08% (*p* < 0.05) compared with that in the APPM group (Figure 2a). As compared with the SCK group, the F_v_/F_0_ (potential activity of photosystem II photochemistry), F_v_/F_m_ (maximum photochemical efficiency of photosystem II), and qP (photochemical quenching coefficient) of *B. napus* leaves in the SPPM group increased by 49.91%, 10.30%, and 14.29%, respectively (*p* < 0.05). In comparison to the ACK group, the F_v_/F_0_, F_v_/F_m_, and qP of *B. napus* leaves in the APPM group increased by 183.73%, 48.07%, and 57.62%, respectively (*p* < 0.05). The PSII (PSII photochemistry) did not differ among the four groups (Figure 2a).

### 2.3. Effects of PPM Application on Antioxidant Enzyme Activity

A comparison between the ACK and SCK treatments revealed that SOD (superoxide dismutase), POD (peroxidase), and CAT (catalase) activity increased by 53.13%, 54.25%, and 77.10% (*p* < 0.05). The activity of SOD, POD, and CAT in the SPPM group increased by 29.21%, 140.17%, and 198.57%, respectively compared with those in the SCK group (*p* < 0.05), and the activity of SOD, POD, and CAT in the APPM group increased by 28.37%, 70.33%, and 77.87%, respectively compared with those in the ACK group (*p* < 0.05). The content of MDA (malondialdehyde) in the SPPM group decreased by 15.42% compared with that in the APPM group (*p* < 0.05) (Figure 2a). RDA (redundancy analysis) analysis (Figure 2b) showed that the leaf photosynthetic parameters (Tr, Ci, qP, and ΦPSII), antioxidant enzyme activities, and photosynthetic pigment content were significantly positively correlated with stem and leaf fresh weight, and they were negatively correlated with leaf REC, K^+^ and Na^+^ content. Meanwhile, F_v_/F_m_, F_v_/F_0_, and qP were positively correlated with root fresh weight, and negatively correlated with K^+^ and Na^+^ content. However, the MDA content and Gs were negatively correlated with the fresh weight of plant organs and positively correlated with the K^+^ content and REC.

### 2.4. Transcriptomic and Metabonomic Analysis

#### 2.4.1. Determination of Photosynthetic Parameters, Chlorophyll Fluorescence Parameters and Plant Fresh Weight

Principal PCA (component analysis) was performed to assess the correlation of the transcriptomes and metabolomes. The results showed that samples of the SCK and ACK group were separated on PC2 (Principal Component 2), and those of the control (SCK and ACK) and PPM (SMMP and AMMP) groups were separated on PC1 (Principal Component 1) based on the metabolite count (Figure 3a). Samples of the S (SCK and SPPM) and A(ACK and APPM) groups were separated on PC2, and those of the control and PPM groups were separated on PC1 based on the gene counts (Figure 3a). Moreover, there were 83 shared DAMs (differentially accumulated metabolites) and 7 shared DEGs (differentially expressed genes) for the four groups (Figure 3a). The number of DEGs and DAMs in the SPPM and APPM groups were more than those in the SCK and ACK groups. In addition, the up-regulated DEGs were more than the down-regulated ones, while the down-regulated DAMs were more than the up-regulated ones (Figure 3a).

Many DEGs and DAMs for the four groups were enriched in pathways related to energy metabolism (including sulfur metabolism and oxidative phosphorylation) and carbohydrate metabolism. DEGs were also enriched in Nitrogen metabolism, carbon fixation in photosynthetic organisms, Photosynthesis-antenna proteins, and Photosynthesis pathways, and DAMs were also enriched in the amino sugar and nucleotide sugar metabolism, tricarboxylic acid (TCA) cycle, carbon metabolism, and carbon fixation pathways in photosynthetic organisms. The DEGs for the SCK vs. ACK were significantly enriched in Nitrogen metabolism and Photosynthesis -antenna proteins pathways, and those for the SCK vs. SPPM were significantly enriched in the sulfur metabolism pathway. The DEGs for the ACK vs. APPM were significantly enriched in photosynthesis-antenna proteins and photosynthesis pathways, while those for the SPPM vs. APPM were significantly enriched in the citrate cycle pathway (Figure 3b).

#### 2.4.2. PPM’s Impacts on Energy Metabolism and Carbohydrate Metabolism Pathways in *B. napus*

DEGs and DAMs with high photosynthesis-related content were enriched in pathways such as Energy metabolism, Carbohydrate metabolism, Photosynthesis-antenna proteins, and Photosynthesis (Figure 4a). Many genes (*BanA09G0675700ZS*, *BanC04G0172900ZS*, and *BanA09G0666300ZS*) were highly expressed in the four groups, but the expression of genes such as *BanA09G0675700ZS*, *BanC04G0172900ZS*, and *BanA09G0666300ZS* were at a low level. Compared with the CK (ACK and SCK), the expression of *BanC02G0540200ZS* and *BanC02G0555600ZS* in the PPM groups (APPM and SPPM) were up-regulated, while those of *BanA03G0578600ZS*, *BanC01G0424400ZS*, and *BanC07G0195800ZS* were down-regulated. Different treatments had different mechanisms in regulating the carbon metabolism pathway (Figure 4a). The expression of *ALDO* (aldolase A) and *FBP* were significantly down-regulated in the Calvin cycle, while that of *RPIA* (ribose 5-phosphate isomerase A) was significantly up-regulated. the expression of *ACLY* (ATP citrate lyase), *FH* (fumarate hydratase), and *SDHB* (Succinate dehydrogenase complex iron sulfur subunit B) were significantly down-regulated in the TCA cycle. Compared with the CK, the relative abundance of metabolites 2-oxoglutric acid and citric acid were significantly down-regulated, while that of fumarate and L-malic acid were significantly up-regulated. The abundance of metabolites related to amino acid metabolism and carbohydrate metabolism was down-regulated.

RDA analysis results (Figure 4b) showed that the expression of photosynthesis-related transcription factors *psb.O* (photosystem II subunit O), *psb.W* (photosystem II subunit W), *psa.O*, and *psa.N* (photosystem I subunit N) and the abundance of fumaric acid, L-malic acid, and 5′-deoxy-5′-(methylthio)adenosine were positively correlated with the fresh weight of plant organs and negatively correlated with REC, K^+^ content, and Na^+^ content. Meanwhile, the expression of *psb.D* (photosystem II subunit D), *psb.W*, *psa.H* (photosystem I subunit H), and *psa.N* and the abundance of fumaric acid, L-malic acid, and 5′-deoxy-5′-(methylthio) adenosine were negatively correlated with the fresh weight of plant organs and positively correlated with REC, K^+^ content, and Na^+^ content.

## 3. Discussion

Irrigation using brackish water with high salt concentration produces ionic toxicity on *B. napus* and inhibits its photosynthesis and cell growth [8]. This study found that the photosynthetic characteristics and antioxidant enzymes activities of *B. napus* showed differences when subjected to different types of brackish water drip irrigation. The leaf photosynthetic parameters (Pn, Tr, and Gs), fluorescence parameters (F_v_/F_0_, F_v_/F_m_ and qP), and Chl b under alkaline water drip irrigation condition were lower than those under neutral water drip irrigation condition. This indicates that alkaline salts have greater inhibition on the leaf photosynthesis of *B. napus*, especially on the net photosynthetic rate and fluorescence parameters. This ultimately inhibits the dry matter accumulation in *B. napus* [25,26].

Long-term brackish water drip irrigation could also lead to massive accumulation of Na^+^ and other salt ions in plants, disrupting osmotic balance, and competitively inhibiting the uptake of other ions [27]. Neutral brackish water irrigation is dominated by Na^+^ and Cl^−^ stress, while alkaline brackish water irrigation also includes CO_3_^2−^, and high pH stress [28,29]. This study found that more Na^+^ and less K^+^ were accumulated in leaves under alkaline water drip irrigation compared to neutral water drip irrigation. A high Na^+^ content could cause high osmotic pressure in *B. napus*, resulting in the closing of leaf stomata and inhibited photosynthesis, metabolism, and biomass accumulation. Besides, excessive Na^+^ can also cause a deficiency of phosphorus and nitrogen, restricting the synthesis of chloroplasts [30,31]. The less K^+^ content under alkaline water drip irrigation may hinder starch synthesis in *B. napus* and accelerate the decomposition of starch into sugar. However, Carbohydrates in leaves cannot be transported smoothly, which causes a massive accumulation of photosynthesis-synthesized sugars in cells. This ultimately causes feedback inhibition and a reduced accumulation of dry matter by photosynthesis. This study also found that alkaline water irrigation had a greater effect on the chlorophyll content of *B. napus* than neutral water irrigation. Chloroplasts are the main sites of plant photosynthesis [32,33]. So, in this study, alkaline water irrigation led to reduced leaf photosynthates and plant biomass.

In this study, the application of PPM regulated *B. napus* growth and photosynthesis during the full flowering stage under brackish water drip irrigation conditions. Photosynthesis and antioxidant defense play key roles in the response of *B. napus* to salt stress [34]. The present study showed that PPM application reduced the ionic toxicity and oxidative stress to *B. napus* under brackish water drip irrigation conditions. This may be because the acidic functional groups such as COOH-, C=O, -SH, and -CHO in PPM could combine with the salt ions brought by brackish water drip irrigation, and improve the structure of soil aggregates and soil structure [35]. This could reduce the uptake of salt ions by *B. napus*, alleviate ionic toxicity and oxidative stress, and ensure normal photosynthesis and biomass accumulation. PPM can also regulate the osmotic potential in *B. napus* leaves by regulating the accumulation of compatible solutes such as soluble sugars, organic acids, amino acids, choline, and betaine, which could protect the photosynthase system [36], thus improving the photosynthetic rate and antioxidant defense system of *B. napus* [37]. In addition, our study found an increase in the MDA content of leaves after PPM application, which may act as one of the adaptive responses of the plant to environmental changes, helping the plant survive under adversity and adapt to changing environmental conditions [38]. Therefore, PPM can maintain the intracellular stability and normal physiological activities of *B. napus* leaves under salt stress, and protect photosynthesis and other physiological processes from serious impact.

*B. napus* can adapt to high salinity environments by molecular and metabolic regulations. The differences in *B. napus*’s self-regulation under different types of brackish water drip irrigation were reflected in pathways related to nitrogen metabolism and photosynthesis. This study (Figure 5a,b) found that under neutral water irrigation, transcription factors *psaN* and *atpF*(ATP synthase F subunit) and metabolites malic acid, glycolyneural, and fumaric were the main responders in the photosynthesis-related pathways and played positive regulatory roles. A previous study [23] has shown that salt stress could inhibit the transcription and translation of psbA (photosystem II reaction center protein D1) encoding D1 protein and the expression of light-induced genes, so that these genes could not be transcribed, translated, and repaired to form an active PSII reaction center. However, the expression of *psbW* (photosystem II reaction center protein W) and *psbN* (photosystem II reaction center protein N) were negatively correlated with the root fresh weight, leaf fresh weight, and the abundance of oxoglutaric acid, and the leaf and root fresh weight and photosynthetic parameters were negatively regulated by Chla, F_v_/F_0_, and oxoglutaric abundance. This indicates that *psbW* and *psbN* could regulate plant growth and photosynthesis under neutral water irrigation conditions. In addition, in the ACK group, transcription factors *psbW* and *atpF* and the metabolites lactone, malic, and fumaric acid played a positive regulatory role on leaf and root fresh weight. In addition, the PSII and Chl a + Chl b played a positive regulatory role on the internal homeostasis of *B. napus* leaves. This indicates that these transcription factors and metabolites directly regulated the growth of *B. napus* under alkaline water irrigation condition. Therefore, under different brackish water irrigation conditions, rapeseed can change the accumulation of metabolites in photosynthesis-related metabolic pathways by regulating the expression of photosynthesis-related genes and metabolites in rapeseed leaves, thereby maintaining the normal growth of plants in adverse environments. It should be noted that compared to neutral water irrigation, alkaline water irrigation had a greater impact on plant cell homeostasis.

It was found that soil amendment could improve plant photosynthetic performance and biomass accumulation by changing the expression of genes, transcription factors and metabolites related to photosynthesis-related pathways under salt stress conditions (Figure 4). However, the application of PPM could regulate transcription factors and metabolites in pathways related to energy metabolism and carbohydrate metabolism in *B. napus* leaves to regulate photosynthates. Under neutral water irrigation, the application of PPM mainly regulated sulfur metabolism, carbon fixation pathways in photosynthetic organisms, and carbohydrate metabolism pathways. The regulatory networks (Figure 5c,d) showed that the application of PPM changed the internal homeostasis of *B. napus* leaves under brackish water drip irrigation, and *atpF*, *psbO*, *psaN*, *psaH*, malic, adenosine, and lactone were the main regulators. It has been reported that *PsbO* (photosystem II oxygen-evolving enhancer protein), *PsbP* (photosystem II oxygen-evolving enhancer protein P), and *PsbQ* (photosystem II oxygen-evolving enhancer protein Q) play a crucial role in maintaining the active site of PSII [39]. This indicates that in this study, PPM may improve the photosynthetic efficiency by maintaining the active site in PSII, thereby increasing the biomass. It was also found that under neutral water irrigation conditions, *atpF* and *psaH* mainly regulated the dry matter. Among them, *atpF* positively regulated the fresh weight and K^+^/Na^+^ ratio, while *psaH* negatively regulated them (Figure 5). *PsbN* and *psbO* mainly regulated the Pn, Tr, POD activity, SOD activity, and MDA content, while *psbN* had an opposite effect on these parameters. Under alkaline water irrigation, the photosynthesis, oxidative phosphorylation, and TCA cycle metabolic pathways were mainly regulated. The regulatory mechanisms of *atpF*, *psaH*, and *psbN* under alkaline water irrigation conditions were similar to those under neutral water irrigation conditions, but *psbO* did not play a good regulatory role under alkaline water irrigation conditions.

This study found that brackish water drip irrigation mainly led to changes in nitrogen metabolism, sulfur metabolism, and oxidative phosphorylation pathways in *B. napus* leaves at the full flowering stage. This is different from the self-regulation mechanism of *B. napus* under salt stress at the seedling stage. Wang reported that under salt stress, the glucose metabolism, amino acid metabolism, and glycerol metabolism pathways were mainly regulated in *B. napus* at the seedling stage [40]. In this study, the application of PPM further changed the regulatory pathways of *B. napus* in the full flowering stage, with Calvin cycle and TCA cycle metabolism as the main metabolic pathways, and enhanced the regulatory effects of transcription factors (*atpF*, *psbO*) and metabolites (fumarate, citric acid).In addition, after the application of PPM, changes in transcription factors (such as *ALDO* (aldolase A), *RPIA* (ribose 5-phosphate isomerase A), and FH (fumarate hydratase)) and metabolites (such as fumarate and, citric acid, ) that were positively correlated with photosynthesis were more under neutral water drip irrigation than under alkaline water drip irrigation. This indicates that PPM has a better effect on alleviating neutral salt stress. It should be noted that under neutral and alkaline water irrigation conditions, the regulatory mechanism of *B. napus* showed great differences (Figure 5), the main factors were inapparent, and the regulatory effects of transcription factors (*atpF*, *psaN*) and metabolites (adenosine, malic) related to photosynthesis were weak. However, after the application of PPM, the regulatory networks under neutral and alkaline water irrigation conditions tended to be consistent. That is, the regulatory effects of transcription factors (*atpF*, *psbO*, *psaH* and *psaN*) and metabolites (malic and adenosine) were enhanced, and the biomass and physiological indicators of *B. napus* were positively regulated.

## 4. Materials and Methods

### 4.1. Experimental Site and Materials

This experiment was conducted in Shihezi, Xinjiang, China (44°32′44.6″ N, 85°99′877″ E). The region is a continental dry climate, with annual sunshine hours of 2300–2700 h, annual average rainfall of 220 mm and annual evaporation of 1000–1500 mm. Soil amendment (PPM), a liquid mixture of polyacrylamide, polyvinyl alcohol, and manganese sulfate prepared at 90 °C was used in this experiment. The PPM material was invented by our research team in 2020. Before application (22 July 2020), the PPM was mixed with inorganic fertilizer, using the following as the mass ratio of the PPM-type modifier:polyvinyl alcohol:polyacrylamide:manganese sulfate:inorganic fertilizer = 1:3:6:50.

### 4.2. Experimental Design

On 15 June 2020, 120 kg of soil (0–60 cm soil layer, pH: 8.25, cation exchange capacity: 17.32 cmol·kg^−1^; alkaline 12ydrolysable nitrogen content: 56 mg·kg^−1^; available phosphorus content: 10.7 mg·kg^−1^; available potassium content: 226 mg·kg^−1^) was collected from the experimental site. Then, the soil was transferred into a cylindrical barrel (0.3 m × 0.6 m × 0.6 m) and the original soil layers were kept. After that, the barrels were buried back to the field. This experiment adopted a randomized complete block design with four groups, namely (1) SCK group (no soil amendment and water containing 6 g·L^−1^ NaCl (neutral salt) was used for drip irrigation), (2) ACK group (no soil amendment and water containing 6 g·L^−1^ Na_2_CO_3_ (alkaline salt) was used for drip irrigation); (3) SPPM group (water containing 6 g·L^−1^ NaCl was used for drip irrigation and 12 g·L^−1^ of PPM was applied); (4) APPM treatment (water containing 6 g·L^−1^ Na_2_CO_3_ was used for drip irrigation and 12 g·L^−1^ of PPM was applied). The pH values of the irrigation solutions under different treatments of ACK, SCK, APPM and SPPM were 7.05, 11.04, 6.89 and 10.53, respectively. Each group had three replicates.

*B. napus* seeds (variety Huayouza 82) were sown after mixing with triple superphosphate (1:15) on 15 July 2020. After emergence, six seedlings were retained in each barrel. According to the experimental design, brackish water was irrigated at 10 d intervals throughout the growth period, and PPM was dissolved in irrigation water and applied through the drip irrigation system during the first irrigation. The irrigated soil is 0–40 cm and the irrigation rate is 750 m^3^·ha^−1^. Six leaves were collected from the plants in each group at the full flowering stage (20 October) and, stored in liquid nitrogen.

### 4.3. Measurement Methods

#### 4.3.1. Determination of Photosynthetic Parameters, Chlorophyll Fluorescence Parameters and Plant Fresh Weight

At the full flowering stage (20 October), photosynthetic parameters including the net photosynthesis rate (Pn), transpiration rate (Tr), stomatal conductance (Gs), and intracellular CO_2_ concentration (C_i_) were measured with a Li-6400 portable photosynthesis instrument (LI-COR, Lincoln, Nebraska, USA) [25]. Then, the maximum fluorescence after dark adaptation (F_v_), minimum fluorescence after dark adaptation (F_0_), maximum fluorescence under light (F_m_), and steady-state fluorescence after light adaptation (F_m_′) were determined with a PAM-2100 modulated chlorophyll fluorometer (WALZ, Germany) [41]. Each measurement was repeated four times.
F_v_/F_0_ = (F_m_ − F_0_)/F_0_(1)
F_v_/F_m_ = (F_m_ − F_0_)/F_m_(2)
ΦPSⅡ = (F_m_′ − F_s_)/F_m_′(3)
qP = (F_m_′ − F_s_)/(F_m_′ − F_0_)(4)

On 23 October, three plants were collected from each group, The roots, leaves and stems were weighed after rinsing with distilled water.

#### 4.3.2. Determination of Chlorophyll and Carotenoids in Plant Leaves

Leaves were cut into small pieces. Then, 0.2 g of leaf sample and 20 mL of extract (absolute ethanol: acetone = 1:1) were mixed evenly in a test tube, sealed, and placed in the dark until the sample turned white. After that, the chlorophyll solution was transferred into a cuvette, and the extract was used as a blank. Finally, colorimetry was conducted at 645 nm, 652 nm, 663 nm, and 440 nm [42].

#### 4.3.3. Determination of Leaf Antioxidant Enzyme Activity and Malondialdehyde (MDA) Content

Leaf superoxide dismutase (SOD), peroxidase (POD), catalase (CAT) activities and malondialdehyde (MDA) content were determined by an NBT photochemical reduction method, guaiacol absorbance method, ultraviolet spectrophotometry and thiobarbituric acid method, respectively [43].

#### 4.3.4. Determination of Na^+^, K^+^ Content and Relative Electrical Conductivity in Leaves

The Na^+^ and K^+^ content and REC were determined via a previous method by AP1200 flame spectrophotometer (AP1200, Shanghai, China) and DDSJ-219L conductivity meter [44,45].

#### 4.3.5. Transcriptomic and Metabolomic Assays

First, 12 leaf samples were rapidly stored in liquid nitrogen and stored by Beijing Biomic Biotechnology Co., Ltd. (Beijing, China) for transcriptomic and metabolomic analysis. The transcriptional sequencing platform was Illumina HiSeq, and the metabolome was determined by UPLC-MS/MS [40].

### 4.4. Statistical Analysis

The data were processed by Excel 2016 software. SPSS 23.0 (SPSS Inc., Chicago, IL, USA) was used for one-way variance (ANOVA) analysis (α = 0.05). Figures was drawn using Origin 8.0 (Origin Lab, Northampton, MA, USA). The co-occurrence relationships were visualized using Gephi 0.9.2 (https://Gephi.org/ (accessed on: 20 March 2022)). Fastp was used to remove low-quality sequences in adapters and reads [46]. The filtered reads were aligned to the reference genome of *B. napus* (https://www.cottongen.org/species/Gossypium_hirsutum/ZJU-AD1_v2.1 (accessed on: 15 December 2021)). The gene expression level was calculated in units of FPKM (Fragments Per Kilo bases Per Million Fragments Mapped). Differential analysis of gene expression was performed using the DESeq package in R software, and differentially expressed genes (DEGs) were selected based on |log2FoldChange| > 1 and *p* < 0.05. Differentially expressed genes and transcription factors were subjected to GO and KEGG enrichment analysis using the clusterProfiler package in R software [47].

## 5. Conclusions

Irrigation using different types of brackish water cause different ionic toxicity to *B. napus*, resulting in differences in photosynthetic characteristics. Alkaline water drip irrigation inhibited the photosynthesis of *B. napus* greater, causing lower photosynthetic (Pn, Tr, and Gs) and fluorescence (F_v_/F_0_, F_v_/F_m_ and qP) parameters and chlorophyll b content in *B. napus* leaves. In addition, there were differences in the expression of photosynthesis-related genes, transcription factors, and metabolites under different types of brackish water drip irrigation conditions, leading to differences in the photosynthesis-related metabolic pathways and photosynthate content. The pathways related to nitrogen metabolism and photosynthesis were changed by brackish water drip irrigation However, the application of PPM reduced the ion toxicity and oxidative stress induced by brackish water drip irrigation, increased the photosynthetic pigment content and improved the chlorophyll fluorescence parameters, thereby enhancing the photosynthetic performance and biomass accumulation. The integrated transcriptomic and metabonomic analysis results showed that the application of PPM could regulate transcription factors and metabolites related to energy metabolism and carbohydrate metabolism-related pathways, to regulate photosynthates and increase biomass. However, these transcription factors and metabolites were different under alkaline and neutral water drip irrigation conditions. Under neutral water irrigation, the application of PPM mainly regulated sulfur metabolism, carbon fixation pathways in photosynthetic organisms, and carbohydrate metabolism pathways in *B. napus* leaves. However, under alkaline water irrigation, the photosynthesis, oxidative phosphorylation, and tricarboxylic acid (TCA) cycle metabolic pathways were mainly regulated (Figure 6). This study revealed the regulation mechanism of PPM on oilseed rape photosynthesis under different brackish water irrigation conditions, and at the same time, it provided a practical method for the use of PPM-type modifiers in agriculture using brackish water irrigation.

## Figures and Tables

**Figure 1 ijms-25-02521-f001:**
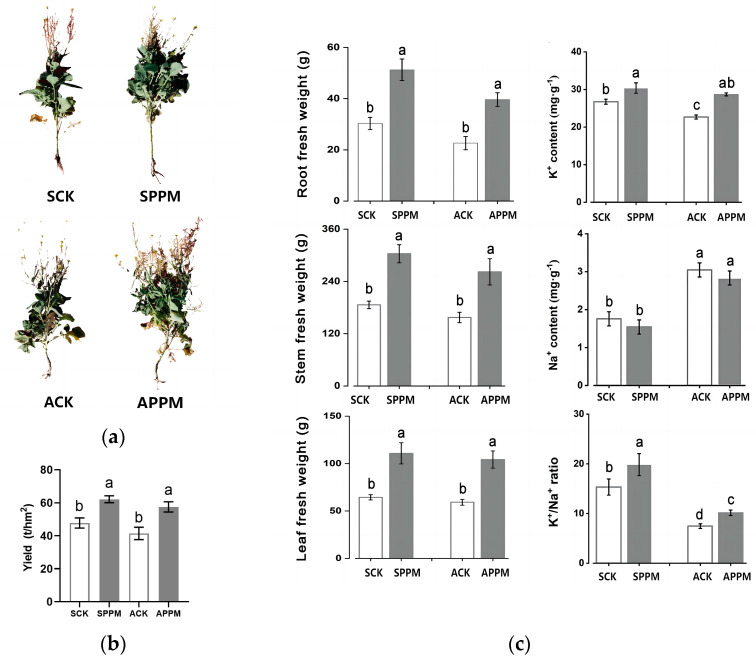
Effects of soil amendment (PPM) on the morphology (**a**), and yield of *Brassica napus* under different treatments (**b**). Fresh weight, and leaf K^+^ and Na^+^ content of *Brassica napus* under drip irrigation with water containing 6 g·L^−1^ NaCl and 6 g·L^−1^ Na_2_CO_3_ (**c**). Different lowercase letters indicate significant difference between groups at *p* < 0.05.

**Figure 2 ijms-25-02521-f002:**
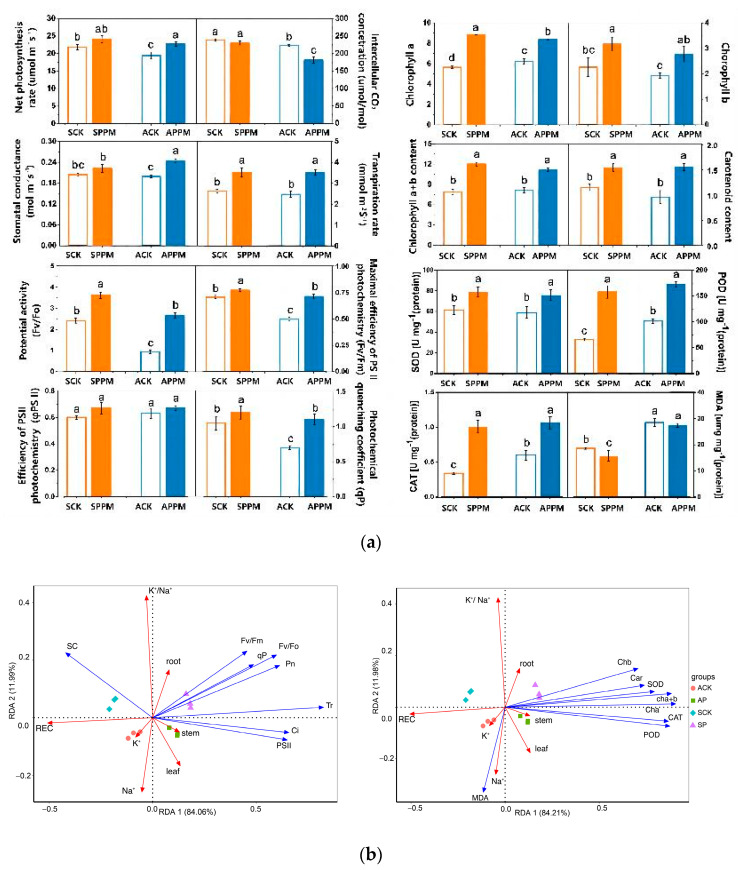
Effects of application of PPM on photosynthetic parameters, pigment content, fluorescence parameters, and antioxidant enzyme activities (**a**) of *Brassica napus* under drip irrigation with waters containing 6 g·L^−1^ NaCl and 6 g·L^−1^ Na_2_CO_3_ and RDA analysis of the parameters. Different lowercase letters indicate significant difference between groups at *p* < 0.05 (**b**).

**Figure 3 ijms-25-02521-f003:**
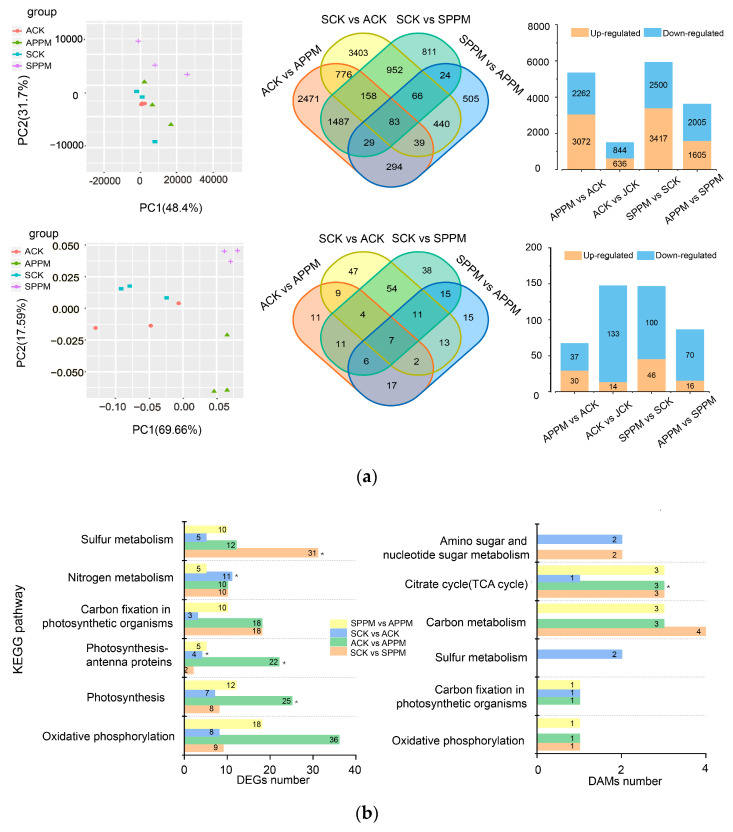
PCA analysis, Venn diagram, up- and down- regulation statistics (**a**), and KEGG annotation of DEGs (differentially expressed genes) and DAMs (differentially accumulated metabolites) in different groups. *, *p* < 0.05 (**b**).

**Figure 4 ijms-25-02521-f004:**
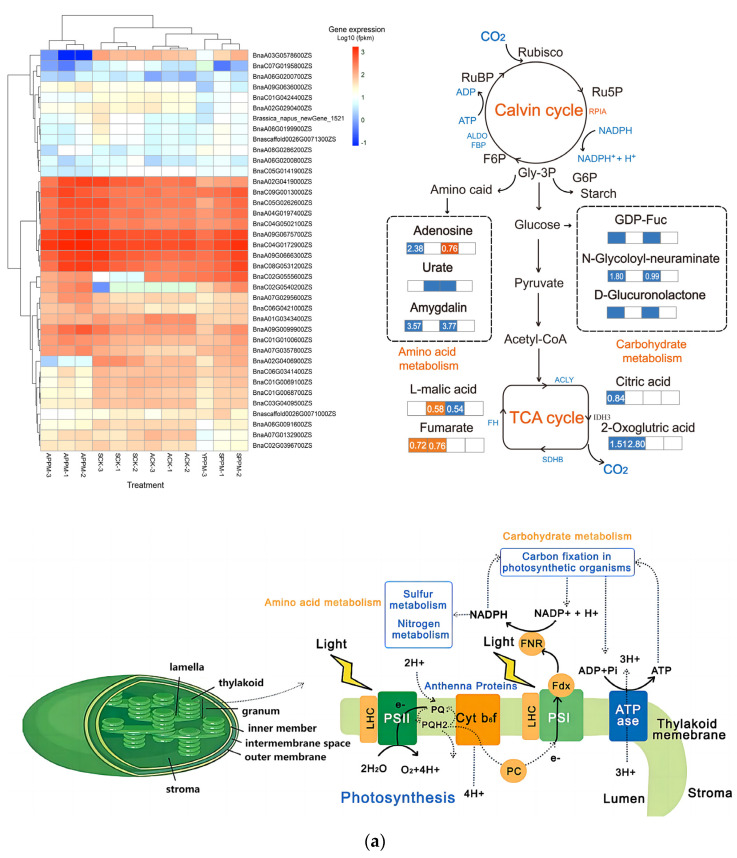
The expression of differentially expressed genes (DEGs) related to photosynthesis in different groups, metabolic pathway analysis, photosynthetic mechanism and energy metabolic pathway analysis (**a**), and RDA analysis of various parameters between groups (**b**). The four boxes in the metabolic pathway represent the ratios of SPPM vs. SCK, APPM vs. ACK, ACK vs. SCK and APPM vs. SPPM metabolite contents, respectively.

**Figure 5 ijms-25-02521-f005:**
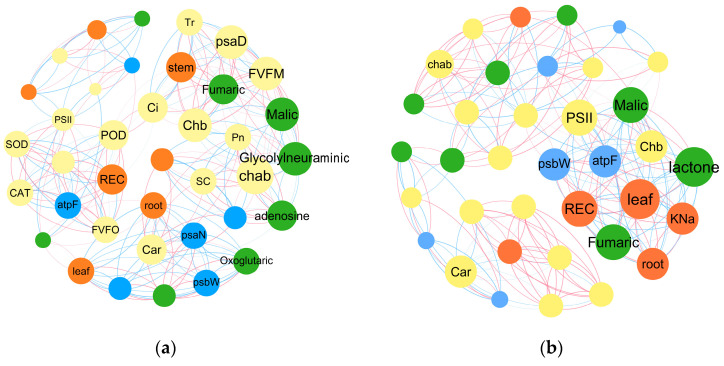
Relationship networks of the yield/ion content (red), photosynthetic parameters (yellow), transcription factors (blue), and DAMs (green) of *B. napus* under different treatments. (**a**) SCK treatment; (**b**) ACK treatment; (**c**) SPPM treatment; and (**d**)APPM treatment. The red lines indicate positive correlation, and the blue lines indicate negative correlation. Node size is positively correlated with the number of connections.

**Figure 6 ijms-25-02521-f006:**
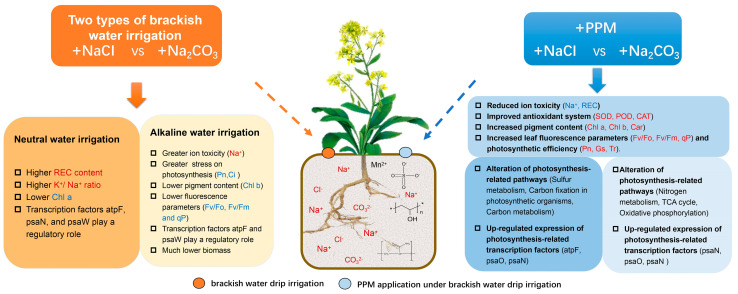
Regulatory mechanism of the effect of soil amendment on rapeseed (*Brassica napus* L.) photosynthesis under drip irrigation with brackish water.

## Data Availability

The raw reads of *Brassica Napus* Transcriptome and Gene expression (TaxID: 3708) are available under accession PRJNA772052 at NCBI Sequence Read Archive (SRA) repository. All data have been released.

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
