# Peer review of "Multi-Omics Analysis of the Effects of Soil Amendment on Rapeseed (Brassica napus L.) Photosynthesis under Drip Irrigation with Brackish Water"

_ijms, 2024, doi:10.3390/ijms25052521_

Round 1

Reviewer 1 Report

Comments and Suggestions for Authors

Thank you for the opportunity to review this paper. I have thoroughly read the manuscrip and I believe the paper is well-written, and the experiment is valid and properly designed and analyzed. This ms must be accepted in present form.

Author Response

Dear reviewers:

Thank you very much for your acknowledgement and support of my manuscript. I am very pleased to hear that you consider my study to be of significant value and agree with the aims and methods of my research. We have further checked and revised the article in details and tried our best to improve the quality of my study.

Once again, I would like to thank you for your review comments and for giving me the opportunity to publish my research in the academic world.

Reviewer 2 Report

Comments and Suggestions for Authors

The article

Multi-omics analysis of the effects of soil amendment on rapeseed (Brassica napus L.) photosynthesis under drip irrigation with brackish water

Ziwei Li, Hua Fan, Le Yang, Shuai Wang, Dashuang Hong, Wenli Cui, Tong Wang, Chunying Wei, Yan Sun, Kaiyong Wang, and Yantao Liu

is aimed at identifying rapeseed response to the use of polymer soil amendment composed of polyacrylamide polyvinyl alcohol and manganese sulfate (PPM). This soil amendment is proposed for usage to improve plants resistance to salinity.

The problem of salinity certainly deserves attention, and the presented work makes a contribution to its solution.

However, when reading the manuscript, a number of questions arise.

1.               First of all, it is necessary to present results of comparison of all the studied parameters for experimental plants with intact control plants that are watered with water without ACK, SCK, APPM, and SPPM; in addition, a control group of plants treated with PPM is required. It is necessary to show that the use of PPM does not change plant metabolism.

2.               Does PPM accumulate in plants?

3.               Please, when first mention in the text of the article, decipher the abbreviations, such as Pn, Gs, Tr, Ci and so on.

 Introduction

4.               Salt stress can also inhibit the carbon assimilation of plants, resulting in reduced accumulation of photosynthates (line 49-50)

Please explain the term photosynthates.

5.               PPM is an independently developed soil amendment with inorganic and organic polymers as the main components (line 69-70) 

By whom and when was РРМ developed?

6.               B. napus is a salt-tolerant plant (line 75-76)

Please provide detailed information on the salt tolerance of rapeseed and compare it with other plants.

 Results

7.               Comparing the ACK group with the SCK group, the root, stem, and leaf fresh weight decreased by 25.37% percent, 25.37%, and 25.37%, respectively (p < 0.05) (line 90-91)

It is not clear in comparison with what object the described changes occur. It is necessary to provide data for control plants.

8.               A comparison between the ACK and SCK treatments revealed that SOD, POD, and CAT activity increased by 53.13%, 54.25%, and 77.10% (p < 0.05). The activity of SOD, POD, and CAT in the SPPM group increased by 29.21%, 140.17%, and 198.57%, respectively compared with those in the SCK group (p < 0.05), and the activity of SOD, POD, and CAT in the APPM group increased by 28.37%, 70.33%, and 77.87%, respectively compared with those in the ACK group (p < 0.05). The content of MDA in the SPPM group decreased by 15.42% compared with that in the APPM group (p < 0.05) (Fig. 2d) (line 132-138)

There is no reference in this text to figure showing these results. In Figure 2, which shows the activity of antioxidant enzymes and MDA content, there is no letter d. Please check the description of the results and more accurately compare the ratio of enzyme activity and MDA content in plants of different groups. The plant groups are probably labeled incorrectly in the figure. Please check.

9.               Principal component analysis (PCA) was performed to assess the correlation of the transcriptomes and metabolomes. The results showed that samples of the SCK and ACK group were separated on PC2, and those of the control (SCK and ACK) and PPM (SMMP and AMMP) groups were separated on PC1 based on the metabolite count (Fig. 3A). Samples of the NaCl and Na2CO3 groups were separated on PC2, and those of the control and PPM groups were separated on PC1 based on the gene counts (Fig. 3D). Moreover, there were 83 shared DAMs and 7 shared DEGs for the four groups (Fig. 3B, E). The number of DEGs and DAMs in the SPPM and APPM groups were more than those in the SCK and ACK groups. In addition, the up-regulated DEGs were more than the down-regulated, 161 while the down-regulated DAMs were more than the up-regulated (Fig. 3C, F) (line 153-161)

This text describes the results shown in Figure 3. However, there are no letters C, D, E, F. Please correct. Also check that the results are described properly. There are a lot of repetitions in the text. It is worth explaining the abbreviations DAM and DEG.

10.            Compared with the CK, the relative abundance of metabolites 2-Oxoglutric acid and Citric acid were significantly down-regulated, while that of Fumarate and L-malic acid were significantly up-regulated. The abundance of metabolites related to Amino acid metabolism and Carbohydrate metabolism were down regulated (line 192-195)

Please point out where these results are given, in what figure or table. If it is Figure 4, in what units the numbers are given?

11.            Samples of the NaCl and Na2CO3 groups were separated on PC2, and those of the control and PPM groups were separated on PC1 based on the gene counts (Fig. 3D) (line 157-158)

Please use the abbreviations of the groups, which are accepted throughout the text of the article.

12.            What are the genes mentioned in the Results section responsible for: BanA09G0675700ZS, 182 BanC04G0172900ZS, BanA09G0666300ZS and others (line 182-188)?

13.            Explain the abbreviations СК, ALDO, FBP, RPIA, ACLY, FH, SDHB and others.

 Discussion

14.            Why are two types of salts compared in the work: the neutral one (NaCl) and the alkaline one (Na2CO3)? In what cases does salinity with these salts occur?

15.            The authors claim that

The present study showed that PPM application reduced the ionic toxicity and oxidative stress to B. napus under brackish water drip irrigation conditions (line 238-239)

However, in Figure 2, MDA content in groups with PPM increased significantly, which indicates severe oxidative damage in plants. Please check whether the groups in the figure are labeled correctly.

 Materials and Methods

16.            What was pH of АСК, SCK, APPM, and SPPM solutions?

 Conclusions

17.            This study provides technical guidance for the use of brackish water in agriculture by the application of polymer soil amendments in arid areas (line 425-426)

What technical guidance is there provided exactly? Please clarify.

Author Response

Dear reviewer:

We sincerely thank the reviewers for thoroughly examining our manuscript and providing very helpful comments to guide our revision, which has significantly improved the presentation of our manuscript. We have carefully considered all comments from the reviewers and revised our manuscript accordingly. We believe that our responses have well addressed all concerns from the reviewers. We hope our revised manuscript can be accepted for publication.

Q1. First of all, it is necessary to present results of comparison of all the studied parameters for experimental plants with intact control plants that are watered with water without ACK, SCK, APPM, and SPPM; in addition, a control group of plants treated with PPM is required. It is necessary to show that the use of PPM does not change plant metabolism.

Response:Thank you for your suggestion. In our preliminary work, we used three soil conditioners with different compositions in field trials on neutral and alkaline saline irrigation, and the effects on the biomass of various organs of oilseed rape at different times are shown in Fig. 1:

Figure1. Changes in the yield of Brassica napus

Based on the experimental results, we selected the amendment treatment with the optimal effect on oilseed rape biomass for further molecular experiments. Due to the experimental conditions and financial constraints we targeted our study to simulate the differences in the effect of one amendment AP2 treatment (PPM-type polymeric amendment) on the photosynthetic mechanism of fodder rape under two conditions of brackish water irrigation and therefore did not set up a blank control group and a plant control group for the PPM treatment. Your suggestions are very useful for us and we will optimise them further in future experiments.

Q2. Does PPM accumulate in plants?

Response:Thank you for your suggestion, the modifier(PPM) we used in the test is a mixture of organic and inorganic compounds, where the polyacrylamide and polyvinyl alcohol used are widely used as flocculants for water body treatment, pharmaceutical aspects and soil conditioners among others (Xiong et al, 2018). In terms of water applications, Sean et al. (2017) showed that when used as a polyacrylamide flocculant, the polyacrylamide compounds tested were not acutely toxic to the mussel species and life cycles tested, suggesting that the use of polyacrylamide in the environment poses minimal risk of short-term exposure. The use in soil conditioning is mainly in improving crop yield and physiological indicators under abiotic stress (Wang et al., 2020; An et al., 2021). Currently, there is no literature on PPM in terms of plant accumulation, and some literature suggests that synthetic polymer conditioners such as PAM are more effective than organic or inorganic conditioners in enhancing soil stability, infiltration rate and erosion resistance, and that their monomers are toxic after polyacrylamide decomposition in the soil, but with low residual levels. Therefore there is uncertainty (Xiong et al, 2018).

 In this study, the amount of polyacrylamide and polyvinyl alcohol in the PPM-type amendment was low, and most of them were immobilised in the soil. The effect of its action was mainly verified in the results of the team's previous research by improving the soil structure and buffering and fixing the salt-based ions in the soil. In addition, similar experiments were conducted in crops such as cotton (An et al., 2021) and wheat, all of which showed that PPM-type amendments were effective in improving yield and physiological indicators of crops under biotic stress. Therefore, most of the indications are beneficial for agricultural production, while our article is to illustrate the effects and molecular mechanisms of improving crop resistance and enhancing their photosynthesis under brackish irrigation to address long-term saline irrigation conditions. At the same time after reviewing the literature we found that polyacrylamide and polyvinyl alcohol have a slow turnover and decomposition rate in soil, and only under high temperature and pyrolysis conditions will be degraded in large quantities, and therefore under natural conditions slower in different soil textures the decomposition rate varies, the cycle is about several years (Nivens et al. 1989; Kolvenbach et al., 2018). We hypothesise that short-term applications of PPM-based amendments to soils in arid zones are largely unlikely to lead to enrichment of PPM in plants. Meanwhile, we have added accordingly in the Introduction section. (On page 3, line 102-105)

Add references:

Nivens, D. E., Ogden,26; Bezdicek, (1989). Decomposition of polyacrylamide in soil columns under laboratory conditions. Soil Science Society of America Journal, 53(4), 1139-1144.

Kolvenbach, B.A., S. Fournier, Q. Mu, and E.M. Bruns. (2018). Polyvinyl alcohol biodegradation in wastewater treatment plants: A review. Water Research, 139, 118-128.

An M, Chang D, Hong D, Fan H, Wang K. Metabolic regulation in soil microbial succession and niche differentiation by the polymer amendment under cadmium stress. J Hazard Mater. 2021 Aug 15;416:126094.

Xiong, B., Loss, R.D., Shields, D.et al.Polyacrylamide degradation and its implications in environmental systems. npj Clean Water 1, 17 (2018).

Buczek SB, Cope WG, McLaughlin RA, Kwak TJ. Acute toxicity of polyacrylamide flocculants to early life stages of freshwater mussels. Environ Toxicol Chem. 2017 Oct;36(10):2715-2721. doi: 10.1002/etc.3821. Epub 2017 Jun 23. PMID: 28397985.

Q3. Please, when first mention in the text of the article, decipher the abbreviations, such as Pn, Gs, Tr, Ci and so on.

Response:Thank you for your suggestion, we have added the full names of Pn, Gs, Tr and Ci in the text and added the Abbreviations section at the beginning of the article.

Where Pn: Leaf net photosynthetic rate; Gs: stomatal conductance; Tr: transpiration rate; Ci: intracellular CO2 concentration. (On page 1, line 32-61)

Introduction

Q4. Salt stress can also inhibit the carbon assimilation of plants, resulting in reduced accumulation of photosynthates (line 49-50)

Please explain the term photosynthates.

Response:Thank you for your suggestion. The term "photosynthates" is interpreted as the substances produced and accumulated during photosynthesis, which are photosynthetic products.

Q5. PPM is an independently developed soil amendment with inorganic and organic polymers as the main components (line 69-70)

By whom and when was PPM developed?

Response:Thank you for your suggestion, PPM was developed together by this scientific team, the developers including the author of this article. It was developed in 2020. We have added to the test material accordingly (On page 13, line 394-395).

Q6. B. napus is a salt-tolerant plant (line 75-76)

Please provide detailed information on the salt tolerance of rapeseed and compare it with other plants.

Response:Thank you for your suggestion, we will provide the specific information of the oilseed rape variety Huayu miscellaneous 82 as follows: Huayu miscellaneous 82, provided by Huazhong Agricultural University, variety source: Zheyou 50×12-4925, belongs to the semi-winter kale-type oilseed rape hybrids. Registration number: GPD Oilseed Rape (2020) 420104. compared to salt-tolerant crops such as cotton and wheat, which resist salt damage by "salt avoidance" (avoiding salt uptake), variety Huayouza 82 is able to take up a large amount of sodium ions and store them in the stalks and petioles. wang et al (2020) found that variety Huayouza 82 was able to grow at salt concentrations of 0.8-1.2% through culture tests with different salt concentrations. Yan (2021) et al. found that variety Huayouza under salt stress at 0.5%-1.5% NaC l concentrations compared to varieties such as Zheshuang72, Zheyou50, Zheda622, and Feeding Oil No. 2 The salinity tolerance performance was the best. In summary, the Huayu heterozygous varieties of oilseed rape possessed strong salt tolerance and were able to grow under high salt environments and produce high yields.

Finally, we added the related research progress in the Introduction section (On page 3, line 106-112).

Add references:

Shah AN, Tanveer M, Abbas A, et al. Targeting salt stress coping mechanisms for stress tolerance in Brassica: A research perspective. Plant Physiol Biochem. 2021 Jan;158:53-64.

Wang W, Pang J, Zhang F, et al. Integrated transcriptomics and metabolomics analysis to characterize alkali stress responses in canola (Brassica napus L.). Plant Physiol Biochem. 2021 Sep;166:605-620.

Yan ZZ, Hu JY, Wu GQ. Salt tolerance test of five oilseed rape varieties and application to saline-alkaline land improvement[J]. Zhejiang Agricultural Science,2021,62(12):2407-2409.

Q7.Comparing the ACK group with the SCK group, the root, stem, and leaf fresh weight decreased by 25.37% percent, 25.37%, and 25.37%, respectively (p < 0.05) (line 90-91)

It is not clear in comparison with what object the described changes occur. It is necessary to provide data for control plants.

Response: Thank you for your suggestion, we are sorry that we did not clearly indicate the comparative relationship in the text, this paragraph is the fresh weight of roots, stems and leaves in the ACK group compared to the SCK group, this sentence should be changed in result section (On page 4 , line128-140). We provided the corresponding data as shown in the table below:

treatment

Yield

 (t/hm2)

Root fresh

weight(g)

stem fresh

weight(g)

leaf fresh weight(g)

ACK

41.41±3.273b

24.76±3.63b

168.25±11.23b

61.74±4.09b

SCK

47.73±4.63b

30.15±2.76b

185.57±9.03b

63.56±4.32b

APPM

57.49±4.78a

42.35±6.53a

283.63±13.65a

107.34±10.34a

SPPM

62.14±3.98a

49.56±8.74a

299.89±9.42a

113.76±9.47a

Q8. A comparison between the ACK and SCK treatments revealed that SOD, POD, and CAT activity increased by 53.13%, 54.25%, and 77.10% (p < 0.05). The activity of SOD, POD, and CAT in the SPPM group increased by 29.21%, 140.17%, and 198.57%, respectively compared with those in the SCK group (p < 0.05), and the activity of SOD, POD, and CAT in the APPM group increased by 28.37%, 70.33%, and 77.87%, respectively compared with those in the ACK group (p < 0.05). The content of MDA in the SPPM group decreased by 15.42% compared with that in the APPM group (p < 0.05) (Fig. 2d) (line 132-138)

There is no reference in this text to figure showing these results. In Figure 2, which shows the activity of antioxidant enzymes and MDA content, there is no letter d. Please check the description of the results and more accurately compare the ratio of enzyme activity and MDA content in plants of different groups. The plant groups are probably labeled incorrectly in the figure. Please check.

Response: Thank you very much for your suggestion, and we apologise for these errors due to an oversight on our part, as the expression is presented in (Figure 2a). The words "(Figure 2d)" should be replaced by "(Figure 2a)". We have corrected it in the original text (On page 5 , line166,171,180 ).

Q9. Principal component analysis (PCA) was performed to assess the correlation of the transcriptomes and metabolomes. The results showed that samples of the SCK and ACK group were separated on PC2, and those of the control (SCK and ACK) and PPM (SMMP and AMMP) groups were separated on PC1 based on the metabolite count (Fig. 3A). Samples of the NaCl and Na2CO3 groups were separated on PC2, and those of the control and PPM groups were separated on PC1 based on the gene counts (Fig. 3D). Moreover, there were 83 shared DAMs and 7 shared DEGs for the four groups (Fig. 3B, E). The number of DEGs and DAMs in the SPPM and APPM groups were more than those in the SCK and ACK groups. In addition, the up-regulated DEGs were more than the down-regulated, 161 while the down-regulated DAMs were more than the up-regulated (Fig. 3C, F) (line 153-161)

Response: Thank you very much for your suggestion, and we apologise for these errors due to our negligence, the above expressions are presented in (Fig. 3a) (On page1-2 , line 32-61). And the expression of this sentence is refined. Meanwhile, the abbreviations of DAM and DEG are: DEGs: differentially expressed genes; DAMs: differentially accumulated metabolites. we have corrected them in the original text.

Q10. Compared with the CK, the relative abundance of metabolites 2-Oxoglutric acid and Citric acid were significantly down-regulated, while that of Fumarate and L-malic acid were significantly up-regulated. The abundance of metabolites related to Amino acid metabolism and Carbohydrate metabolism were down regulated (line 192-195)

Please point out where these results are given, in what figure or table. If it is Figure 4, in what units the numbers are given?

Response: These results are derived from the analysis of Figure 4A, where the numerical units are the ratios under different treatments, where the first number is SPPM vs SCK, the second number is APPM vs ACK, the third number is ACK vs SCK, and the fourth is APPM vs SPPM (On page 9, line 252-254 ).

Q11. Samples of the NaCl and Na2CO3 groups were separated on PC2, and those of the control and PPM groups were separated on PC1 based on the gene counts (Fig. 3D) (line 157-158)

Please use the abbreviations of the groups, which are accepted throughout the text of the article.

Response: Thank you for your suggestion, and we apologise for not making the article clear. We have changed "NaCl group and Na2CO3 group" to "S group and A group". The abbreviations have been standardised in the article (On page 7 , line 199).

Q12. What are the genes mentioned in the Results section responsible for: BanA09G0675700ZS, 182 BanC04G0172900ZS, BanA09G0666300ZS and others (line 182-188)?

Response: Thank you for your suggestion, the genes mentioned in the results of the article are all differential genes in pathways such as regulation of photosynthetic antenna proteins and photosynthesis. We annotated the specific functions of these genes as follows:BanA09G0675700ZS, BanC04G0172900ZS, and BanA09G0666300ZS are genes that regulate the transcription factor, MSP.1.BanA09G0675700ZS, BanC04G0172900ZS, and BanA09G0666300ZS are genes that regulate psaN. banC02G0540200ZS and BanC02G0555600ZS are genes that regulate the transcription factor, atpF. banA03G0578600ZS, BanC01G0424400ZS, and BanC07G0195800ZS regulate genes for the transcription factor psbO.

Q13. Explain the abbreviations CK, ALDO, FBP, RPIA, ACLY, FH, SDHB and others.

Response:Thank you for your question, we have annotated these abbreviations in the article as follows: CK: control trreatment;.

The full name of the transcription factor ALDO is Aldolase A; the full name of the transcription factor FBP is Fructose-1,6-bisphosphatase;

The full name of transcription factor RPIA is Ribose 5-phosphate isomerase A;

The full name of the transcription factor ACLY is ATP citrate lyase;

The full name of the transcription factor FH is Fumarate hydratase;

The full name of the transcription factor SDHB is Succinate dehydrogenase complex iron sulfur subunit B;

We made additions and corrections in the manuscript accordingly.

Discussion

Q14.Why are two types of salts compared in the work: the neutral one (NaCl) and the alkaline one (Na2CO3)? In what cases does salinity with these salts occur?

Response: Thank you for your question, firstly, freshwater resources are in short supply in arid regions and brackish water is one of the more common and readily available sources of water. The composition of brackish water mainly consists of a variety of salts and minerals dissolved in the water. The main salt components include chlorides, sulphates and carbonates. Specific components and salt concentrations vary depending on geographic location and water source. As in the case of complex saline soil conditions in nature, some soils are soil salinisation caused by enrichment of neutral salts with NaCl as the main salt, and some are caused by enrichment of Na2CO3 as the main ion. However, it is important to note that irrigation with higher concentrations of brackish water may have negative impacts on soil ecosystems, such as soil salinisation and plant acclimatisation problems. Neutral brackish water irrigation is dominated by Na+ and Cl- stress, while alkaline brackish water irrigation also includes CO32-, and high pH stress. The use of salts in irrigation and excessive accumulation of salts in the soil can lead to agricultural salinisation. Although NaCl is the major contributor to soil salinity, Na2CO3 can also contribute to soil alkalisation and salinisation under certain conditions. Therefore, suitable management and control measures should be implemented when saline irrigation is carried out. In this study, we compared the differences in plant response to two brackish water irrigation conditions by adding a chemical polymeric soil conditioner, PPM, which is important for the potential improvement and utilisation of saline soils for agricultural production.

Meanwhile, we have added accordingly in the discussion section of the paper (On page 10,line 266-270).

Add references:

Lu P, Dai SY, Yong LT, et al. A Soybean Sucrose Non-Fermenting Protein Kinase 1 Gene, GmSNF1, Positively Regulates Plant Response to Salt and Salt-Alkali Stress in Transgenic Plants. Int J Mol Sci. 2023 Aug 5;24(15):12482.

Smith, J. D., Johnson, R. L. Saline water irrigation: effects of soil characteristics and irrigation management on crop yield and quality. Agricultural Water Management, 2019, 221, 301-312.

Q15.The authors claim that The present study showed that PPM application reduced the ionic toxicity and oxidative stress to B. napus under brackish water drip irrigation conditions (line 238-239)

However, in Figure 2, MDA content in groups with PPM increased significantly, which indicates severe oxidative damage in plants. Please check whether the groups in the figure are labeled correctly.

Response: Thank you very much for your question. The increase in MDA content is a manifestation of physiological responses made by the plant, indicating that the plant is affected by external pressure or stress, especially oxidative stress. In the article, the MDA content increased after applying PPM treatment, and we speculate that the pre-existing period may be because the plant is actively responding to oxidative stress. We reviewed related studies and found that the increase in plant MDA content also has positive significance to a certain extent, in which 1) activation of antioxidant defence system: the accumulation of MDA can act as a signal molecule that triggers the antioxidant defence system of the plant, prompting the plant to produce more antioxidant substances to counteract oxidative damage, such as superoxide dismutase (SOD), peroxidase (POD) and so forth, which can protect the plant cells from further oxidative damage.2) Promote cell wall increase: the increase of MDA usually leads to lipid peroxidation of cell membranes, which in turn triggers the alteration and increase of cell walls, which helps to enhance the resistance, mechanical strength and resilience of the plant.3) Trigger plant defence responses: the accumulation of MDA may activate the plant's defence responses including the production of antimicrobial substances, anthelmintic substances and hormones, and thus enhance the plant's resistance. and thus enhance plant resistance.4) Promote plant adaptation to the environment: the increase in MDA may serve as one of the plant's adaptive responses to environmental changes, helping the plant to survive under adversity and adapt to changing environmental conditions. In addition, the increase of MDA is only a kind of stress response of plants to external stresses, which can be positive only at moderate levels, and too high levels of MDA may cause damage to plants. Meanwhile we have added a relevant discussion in the discussion section of the article (On page 11, line 398-301).

Add references:

Chen Z, Chen W, Dai F. Study on the physiological response differences of Pinus yunnanensis seedlings under different drought conditions. Journal of Fujian Forestry Science and Technology, 2012, 39(3), 218-222.

Hossain, M. A., Mostofa, M. G., Fujita, M., & Tran, L. S. P. (2014). Physiological and biochemical mechanisms associated with trehalose-induced copper-stress tolerance in rice. Scientific Reports, 4, 1-14.

Sharma, P., & Dubey, R. S. (2005). Modulation of nitrate reductase activity in rice seedlings under aluminium toxicity and water stress: Role of osmolytes as  enzyme protectant. Journal of Plant Physiology, 162(8), 854-864.

Materials and Methods

Q16. What was pH of ACK, SCK, APPM, and SPPM solutions?

Response: Thank you for the suggestion that the pH values of the ACK, SCK, APPM and SPPM treatment solutions in this experiment were 7.05, 11.04, 6.89 and 10.53, respectively.We have added accordingly in Materials and Methods.(On page 14 , line 410-412)

Q17. This study provides technical guidance for the use of brackish water in agriculture by the application of polymer soil amendments in arid areas (line 425-426)

What technical guidance is there provided exactly? Please clarify.

Response: Thank you for your suggestion, the specific guidance in this paper includes providing guidance on application scenarios, application rates and application methods for polymers in brackish water irrigation. Among the application scenarios is the application rate of 12 g/L of polymer modifier PPM under brackish water irrigation conditions with neutral and alkaline salts.We have added accordingly in the conclusion section of the article(On page 16 , line 491-494 ).

In the end, we sincerely hope that this revised manuscript has addressed all your comments and suggestions. We appreciated for reviewers’ warm work earnestly, and hope that the correction will meet with approval. We would like to thank the referee again for taking the time to review our manuscript.

Reviewer 3 Report

Comments and Suggestions for Authors

The paper under the title Multi-omics analysis of the effects of soil amendment on rape seed (Brassica napus L.) photosynthesis under drip irrigation with brackish water written by Ziwei Li and the co-authors is an agriculture technology paper IJMS (ISSN 1422-0067. 

The authors performed an experiment. The Brassica napus plants have been watered with the water solution of 6 g·L-1 NaCl (S) and Na2CO3 (A). In the agricultural experiment, the authors tested the regulation mechanism of polymer soil amendment composed of polyacrylamide polyvinyl alcohol, and manganese sulfate (PPM) on Brassica napus rapeseed photosynthesis under irrigation of the two types of brackish water. 

The authors compared the differences in the effects of drip irrigation with neutral and alkaline water on the physiological and photosynthetic parameters of B. napus, and the differences in the mechanisms of PPM improving photosynthesis of B. napus (seeds (variety Huayouza). This study will have guiding significance for brackish water drip irrigation and crop yield increase in arid areas

The following parameters have been measured in the revised study photosynthetic parameters, chlorophyll fluorescence parameters and plant fresh weight; chlorophyll and carotenoids in plant leaves; leaf antioxidant enzyme activity and malondialdehyde (MDA) content; Na+ , K+ content and relative electrical conductivity in leaves; transcriptomic and metabolomic assays; 

The crucial sentence at the end of the conclusion session „This study provides technical guidance for the use of brackish water in agriculture by the application of polymer soil amendments in arid areas” shows clearly that the paper is a technical guidance.

I am not convinced that there is any science in this paper. There are molecular methods used to assess the differences in plant growth under the two ways of irrigation.

My main concern is related to the impact of the artificial substances polyacrylamide polyvinyl alcohol, and manganese sulfate (PPM) on the soil microbiology. There have been many agricultural technologies that brought a lot of harm to the environment. 

Author Response

Dear reviewer:

We sincerely thank the reviewers for thoroughly examining our manuscript and providing very helpful comments to guide our revision, which has significantly improved the presentation of our manuscript. We have carefully considered all comments from the reviewers and revised our manuscript accordingly. We believe that our responses have well addressed all concerns from the reviewers. We hope our revised manuscript can be accepted for publication.

Q:The crucial sentence at the end of the conclusion session This study provides technical guidance for the use of brackish water in agriculture by the application of polymer soil amendments in arid areas” shows clearly that the paper is a technical guidance.

I am not convinced that there is any science in this paper. There are molecular methods used to assess the differences in plant growth under the two ways of irrigation.

My main concern is related to the impact of the artificial substances polyacrylamide polyvinyl alcohol, and manganese sulfate (PPM) on the soil microbiology. There have been many agricultural technologies that brought a lot of harm to the environment.

Respond:Thank you for your detailed suggestions and recognition of the value of our work, we believe that the scientific significance of this paper is to reveal the mechanism of regulation of photosynthesis in kale-type oilseed rape by PPM-based amendments under two brackish water irrigation conditions, and the practical significance is to provide a practical approach for the use of amendments in agriculture using brackish water irrigation.

From the current status of the application of polyacrylamide polyvinyl alcohol and manganese sulphate (PPM), the improver we used in our experiment is a mixture of organic and inorganic substances, in which the polyacrylamide and polyvinyl alcohol used are widely used as flocculating agents for water treatment, pharmaceutical aspects and soil conditioning agents, etc. (Xiong et al, 2018). In terms of water applications, Sean et al. (2017) showed that when used as a polyacrylamide flocculant, the polyacrylamide compounds tested were not acutely toxic to the mussel species and life cycles tested, suggesting that the use of polyacrylamide in the environment poses minimal risk of short-term exposure. The use in soil conditioning is mainly in improving crop yields and physiological indicators under abiotic stress (Wang et al., 2020; An et al., 2021). It has been shown in the literature that synthetic polymeric conditioners such as PAM are more effective than organic or inorganic conditioners in enhancing soil stability, infiltration rate and erosion resistance, and that the monomers of polyacrylamide are somewhat toxic after decomposition in the soil, but with low residual levels. Therefore there is uncertainty (Xiong et al, 2018). Therefore, the results of most studies indicate that both materials are beneficial for agricultural production . There are no problems such as acute toxicity. Meanwhile, in our previous work, we used three soil conditioners with different compositions in a field trial on neutral and alkaline salt irrigation, and the effects on the yield of each organ of oilseed rape at different times are shown in Fig. 1:

Figure1. Changes in the yield of Brassica napus

Based on the experimental results, we selected the improver treatment with the optimal effect on oilseed rape biomass for further molecular and multi-omics experiments. Specific yield indicators, physiological and biochemical analysis methods to multi-omics analysis at the molecular level were used in the experiments to both verify the ability of PPM-type amendments to increase the yield of kale-type oilseed rape under brackish water irrigation and to reveal the gene and metabolite regulatory pathways associated with the photosynthetic pathway of olive-type oilseed rape when PPM-type amendments were applied to the soil.

From the soil ecological point of view, the amendment we used in the experiment was a mixture of organic and inorganic substances, of which polyacrylamide and polyvinyl alcohol in the PPM-type amendment were used in small amounts and mostly immobilised in the soil. The main effect of its action was verified in the results of the team's previous research by improving the soil structure and buffering the immobilisation of salt-based ions in the soil. In addition, similar experiments in cotton (An et al., 2021) and wheat have shown that PPM-type amendments are effective in improving the yield and physiological indexes of crops under biotic stresses. Meanwhile, after reviewing the literature we found that the turnover and decomposition of polyacrylamide and polyvinyl alcohol in soil is very slow and varies in different soil textures, with a cycle of about several years (Nivens et al., 1989; Kolvenbach et al., 2018). Other results from our team have shown that the application of PPM-type amendments to soils in the face of abiotic stresses affected soil biome activity by increasing the number of endemic soil bacteria (Flaviaesturariibacter, Rubellimicrobium, Cnuella), fungi (Verticillium, Tricharina), actinomycetes (Blastococcus, Nocardioides) and fungus-feeding nematode (Aphelenchus) activities as well as metabolic functions of soil nucleotides and carbohydrates (An, 2022). It was demonstrated that PPM-type amendments have a regulatory effect on soil ecology, and soil microenvironment in adverse environments. Meanwhile, we have added and modified the relevant contents in the article.(On page 3 , line 103-104)

Finally, we revised the conclusion to read "This study reveals the regulation mechanism of PPM on oilseed rape photosynthesis under different brackish water irrigation conditions, and provides a practical method for the use of PPM-type modifiers in agriculture using brackish water irrigation".(On page 16 , line 491-494 )。

Add references:

Nivens, D. E., Ogden,26; Bezdicek, (1989). Decomposition of polyacrylamide in soil columns under laboratory conditions. Soil Science Society of America Journal, 53(4), 1139-1144.

Kolvenbach, B.A., S. Fournier, Q. Mu, and E.M. Bruns. (2018). Polyvinyl alcohol biodegradation in wastewater treatment plants: A review. Water Research, 139, 118-128.

An M, Chang D, Hong D, Fan H, Wang K. Metabolic regulation in soil microbial succession and niche differentiation by the polymer amendment under cadmium stress. J Hazard Mater. 2021 Aug 15;416:126094.

An M, Chang D, Wang X, Wang K. Protective effects of polymer amendment on specific metabolites in soil and cotton leaves under cadmium contamination. Ecotoxicol Environ Saf. 2023 Oct 1;264:115463.

An M, Chang D, Hong D, Fan H, Wang K. Metabolic regulation in soil microbial succession and niche differentiation by the polymer amendment under cadmium stress. J Hazard Mater. 2021 Aug 15;416:126094.

In the end, we sincerely hope that this revised manuscript has addressed all your comments and suggestions. We appreciated for reviewers’ warm work earnestly, and hope that the correction will meet with approval. We would like to thank the referee again for taking the time to review our manuscript.

Reviewer 4 Report

Comments and Suggestions for Authors

The paper "Multi-omics analysis of the effects of soil amendment on rape- 2 seed (Brassica napus L.) photosynthesis under drip irrigation with brackish water" is an original experimental study containing interesting new facts in terms of the mechanisms of photosynthetic activity under the influence of irrigation with brackish water and treatment with soil ameliorants. After making the necessary changes, according to the comments, the article can be published in Int. J. Mol. Sci.

Remarks:

1. The reference list should be carefully checked and supplemented (changed). In particular, the reference to the paper doi: 10.3389/fpls.2021.636536 on lines 73.74 of the manuscript (reference number [17]) is not confirmed by the reviewer. In [17] (doi:10.3389/fpls.2021.636536) there is no data on biodegradation, specific surface area, and aggregate structure of PPM-type composite ameliorants. You are misleading readers by giving such a link. Please find and cite real articles that have studied biodegradation, specific surface area, aggregate structure, etc. for polymer composites with acrylic and/or polyvinyl polymer matrix and give the correct references. In addition, the thesis about the high biodegradability of acrylic polymer ameliorants is very controversial. Synthetic gels are usually classified as non–biodegradable [ doi: 10.1002/app.53655, doi:10.1002/agg2.20074] in contrast to biodegradable biopolymers (polysaccharides) as agents of controlled release systems for agrochemicals [doi: 10.1007/s13593-014-0263-0]. Only a few researchers, for example, Smagin et al [doi:10.3390/ma11101889, doi:10.3390/polym15173582] show in field and laboratory experiments the possibility of rapid biodegradation of PAA and other acrylic gels.

It is necessary to take into account real relevant publications. Or an alternative way is to remove information about biodegradability, specific surface area, aggregate structure and other indicators from the introductory part of the manuscript.

2. The experimental design is not sufficiently described. In particular, it is unclear what the PPM dose of 12 g/l means (lines 346-348)? Per liter of soil? Then in which layer? Or in the entire 0.6m barrel? In what form was it applied (dry or swollen gel), how was it embedded in the soil (mechanical mixture, separate layer?). Or was it generally applied with irrigation water at a dose of 12 g /l of water? Then specify the total volume of irrigation water and the total (cumulative) dose per m2 per season. Further, the proportion of 1:3:6:50 is by weight  or by volume? If by weight it is a dry substance (dry PAA, dry manganese sulfate and organic fertilizer)? It is necessary to clarify everything and give a calculation of the final doses in g/m2, taking into account the area of your barrel. Right here, what does a barrel with dimensions 0,3*0,6*0,6 m (line 340)? How can this be for a cylinder where there are only two parameters – diameter and height?

3. The same is for biomass (Figure 1). It is not correct to express biomass in extensive units (grams). I'll take another barrel (bigger) and get a different weight with the same treatments. For the possibility of comparing experimental data, it is customary to use specific values of biomass per unit area. Please recalculate this weight by the area of your barrel. And, hopefully, you took into account the weight of all 6 plants, not just the selected ones. Otherwise it's not right.

4. Please explain why the differences in biomass (Figure 1) are large (up to 2 times), and the differences in net photosynthesis rate (Figure 2a (upper diagram) are small, and sometimes not reliable at all? Why, with almost the same net photosynthesis rate, did such strong differences in plant biomass occur after PPM treatment? Perhaps a one-time measurement of photosynthesis is not representative?

Summary: The research is interesting, useful and informative. But the manuscript needs to be corrected. Therefore, the reviewer suggests a major revision with a request to take into account all the above comments in the new version of the manuscript.

January 09. 2024.

Best regards, Your Reviewer

Comments on the Quality of English Language

Please check the terms carefully. The name is "Multi-omics analysis..." (maybe a "Multi-system" one is better?), a barrel with three parameters (0,3*0,6*0,6 m) (maybe it was a box, not a barrel?), etc.

Author Response

Dear reviewer:

We sincerely thank the reviewers for thoroughly examining our manuscript and providing very helpful comments to guide our revision, which has significantly improved the presentation of our manuscript. We have carefully considered all comments from the reviewers and revised our manuscript accordingly. We believe that our responses have well addressed all concerns from the reviewers. We hope our revised manuscript can be accepted for publication.

Q1: The reference list should be carefully checked and supplemented (changed). In particular, the reference to the paper doi: 10.3389/fpls.2021.636536 on lines 73.74 of the manuscript (reference number [17]) is not confirmed by the reviewer. In [17] (doi:10.3389/fpls.2021.636536) there is no data on biodegradation, specific surface area, and aggregate structure of PPM-type composite ameliorants. You are misleading readers by giving such a link. Please find and cite real articles that have studied biodegradation, specific surface area, aggregate structure, etc. for polymer composites with acrylic and/or polyvinyl polymer matrix and give the correct references. In addition, the thesis about the high biodegradability of acrylic polymer ameliorants is very controversial. Synthetic gels are usually classified as non–biodegradable [doi: 10.1002/app.53655, doi:10.1002/agg2.20074] in contrast to biodegradable biopolymers (polysaccharides) as agents of controlled release systems for agrochemicals [doi: 10.1007/s13593-014-0263-0]. Only a few researchers, for example, Smagin et al [doi:10.3390/ma11101889, doi:10.3390/polym15173582] show in field and laboratory experiments the possibility of rapid biodegradation of PAA and other acrylic gels.

It is necessary to take into account real relevant publications. Or an alternative way is to remove information about biodegradability, specific surface area, aggregate structure and other indicators from the introductory part of the manuscript.

Respond: Thank you for your suggestion, which is of great help to us. After discussion and reviewing the relevant literature, we decided to delete the information about other indicators such as biodegradability and specific surface area in the introduction according to your suggestion. The effect of its soil amelioration is taken as the main discourse. Among them, the polymeric structure was shown in the previous research of our team, and the references are: An M, Chang D, Hong D, Fan H, Wang K. Metabolic regulation in soil microbial succession and niche differentiation by the polymer amendment under cadmium stress. J Hazard Mater. 2021 Aug 15;416:126094. doi: 10.1016/j.jhazmat.2021.126094. Epub 2021 May 15. PMID: 34492903.

Specific findings are displayed below:

  1. Results

3.1. Fourier Transform Infrared Spectroscopy (FTIR) test analysis

According to the FTIR analysis, the composition of the polymer amendment was complex (Fig. 1). The absorption at 3423 cm−1 and 1636 cm−1 were attributable to the N-H stretching vibration and the variable angle vibration of the primary amine in the polymer amendment, respectively, and the absorption at 1118 cm−1, 1175 cm−1 and 1075 cm−1 were attributable to the C-OH stretching vibration of the secondary hydroxyl, tertiary hydroxyl, and primary hydroxyl, respectively.

Fig. 1. FTIR spectra of the polymer amendment and soil. CK, Cd, PA are soil samples of each treatment; LPA is polymer amendment sample. The data represents relative intensities of major absorption peaks of FTIR spectra of soil under different treatments (Semi-quantitative).

Q2. The experimental design is not sufficiently described. In particular, it is unclear what the PPM dose of 12 g/l means (lines 346-348)? Per liter of soil? Then in which layer? Or in the entire 0.6m barrel? In what form was it applied (dry or swollen gel), how was it embedded in the soil (mechanical mixture, separate layer?). Or was it generally applied with irrigation water at a dose of 12 g /l of water? Then specify the total volume of irrigation water and the total (cumulative) dose per m2 per season. Further, the proportion of 1:3:6:50 is by weight  or by volume? If by weight it is a dry substance (dry PAA, dry manganese sulfate and organic fertilizer)? It is necessary to clarify everything and give a calculation of the final doses in g/m2, taking into account the area of your barrel. Right here, what does a barrel with dimensions 0,3*0,6*0,6 m (line 340)? How can this be for a cylinder where there are only two parameters – diameter and height?

Respond: Thank you for your question, we will answer your question in points:

1) 12 gL-1 indicates the liquid concentration of PPM applied. PPM was compounded out as a liquid form.PPM was dissolved in irrigation water and applied through the drip irrigation system during the first irrigation. The irrigated soil was 0-40 cm and the volume of irrigation water was 750mha-1.(On page 14 , line 417-418)

2) 1:3:6:50 is the mass ratio indicating that polyvinyl alcohol: polyacrylamide: manganese sulphate: inorganic fertiliser is in the mass ratio of 1:3:6:50.(On page 14 , line 396-397)

3) In this study, a cylindrical drum was used, which has a length*width*height of 0.3*0.6*0.6 metres.(On page 14 , line 402)

We have supplemented the Experimental Methods and Materials section of the article accordingly.

Q3. The same is for biomass (Figure 1). It is not correct to express biomass in extensive units (grams). I'll take another barrel (bigger) and get a different weight with the same treatments. For the possibility of comparing experimental data, it is customary to use specific values of biomass per unit area. Please recalculate this weight by the area of your barrel. And, hopefully, you took into account the weight of all 6 plants, not just the selected ones. Otherwise it's not right.

Respond: Thanks to your suggestion, the average of the biomass of the different organs of six oilseed rape plants per bucket was used in this study. We have followed your suggestion and added the average biomass of the plants in each bucket and expressed the biomass as a unit area. The details have been modified in the manuscript.(On page 4 , line 128-145).The modified diagram is shown below:

Q4. Please explain why the differences in biomass (Figure 1) are large (up to 2 times), and the differences in net photosynthesis rate (Figure 2a (upper diagram) are small, and sometimes not reliable at all? Why, with almost the same net photosynthesis rate, did such strong differences in plant biomass occur after PPM treatment? Perhaps a one-time measurement of photosynthesis is not representative?

Respond: Thank you for your suggestion, we are sorry for the trouble caused by our mistake, after checking the data, we found that there is a mistake in the labelling of the data in Fig. 2a, the sequence should be changed from "SCK, ACK, SPPM, APPM" to "SCK, SPPM,ACK, APPM". We have modified the labelling in the figure. We also checked the labelling of other data.The modified diagram is shown below:

The changed results are in the results section of the manuscript. (On page 6 , line 187-191)

In the end, we sincerely hope that this revised manuscript has addressed all your comments and suggestions. We appreciated for reviewers’ warm work earnestly, and hope that the correction will meet with approval. We would like to thank the referee again for taking the time to review our manuscript.

Round 2

Reviewer 2 Report

Comments and Suggestions for Authors

The article has been significantly revised and corrected.At the same time, it is still necessary to provide information about the biological and environmental safety of PPM components in the introduction.Now, with the introduction, only the authors of this article are cited.Please provide the results of other scientific groups working with similar composites.

It is a pity that the groups of plants not exposed to salt and the group exposed to PPM only were not analyzed. This would allow the results presented in this work to be more convincing.

Author Response

Dear reviewer:

We sincerely thank the reviewers for thoroughly examining our manuscript and providing very helpful comments to guide our revision, which has significantly improved the presentation of our manuscript. We believe that our responses have well addressed all concerns from the reviewers. We hope our revised manuscript can be accepted for publication.

Q1:The article has been significantly revised and corrected. At the same time, it is still necessary to provide information about the biological and environmental safety of PPM components in the introduction. Now, with the introduction, only the authors of this article are cited. Please provide the results of other scientific groups working with similar composites.

Response:

Thank you for your suggestion, and we have provided information on the biological and environmental safety of PPM ingredients in the introduction, citing information from other scientific groups working on similar composites. The additions and references are listed below:

Polyacrylamide (PAM), polyvinyl alcohol (PVA) and manganese sulphate are effective as soil conditioners in agriculture to improve soil structure, enhance water retention and reduce erosion. Studies have shown that polyacrylamide has no significant negative effects on aquatic ecosystems and soil organisms when used at recommended doses (10-20 kg·hm-2) (Sojka et al., 2007).The biodegradability of PVA depends on its molecular weight and other structural properties (Thakur and Thakur, 2014). In terms of ecological safety, polyvinyl alcohol is safer, but with increasing decomposition time, there is a need to be concerned about its long-term accumulation in soil and water bodies. Proper use of manganese sulphate avoids manganese deficiency and thus enhances plant growth and resistance, and generally the safe concentration of manganese sulphate solution is 0.1 to 1 per cent (Kabata-Pendias and Mukherjee, 2007). Therefore, their use under recommended conditions is generally considered relatively safe for the environment.

We have added accordingly in the introductory section(On page 3 , line 104-105)

Add references:

Sojka RE, Bjorneberg DL, Entry JA, et al. Polyacrylamides in agriculture and environmental land management. Advances in Agronomy. 2007;92,75-162.

Thakur VK, & Thakur MK. Recent advances in PVA hydrogels, nanocomposites and their biomedical applications. Polymer-Plastics Technology and Engineering. 2014;53(7), 631-645.

Kabata-Pendias, A, & Mukherjee AB.Trace Elements from Soil to Human. Springer. 2007;5:52.

Q2:It is a pity that the groups of plants not exposed to salt and the group exposed to PPM only were not analyzed. This would allow the results presented in this work to be more convincing.

Response:Thank you again for your suggestions, reviewer, we have benefited greatly from your expertise and careful review. At the same time, your suggestions have helped us a lot for the future development of our work and experimental design. I deeply realise that there are still some problems and deficiencies in my thesis, and I will seriously think about them and try to improve them. Your criticisms and suggestions make me more conscious of my own shortcomings, which will motivate me to study harder and improve my academic level.

Thank you again for your support and help. I look forward to receiving your guidance and encouragement again in my future academic path. In the future, we will strengthen the logic and persuasiveness of our article.

Best wishes!

With kind regards.

Ziwei Li

Reviewer 4 Report

Comments and Suggestions for Authors

Dear colleagues, the new version of the manuscript is quite correct. I recommend it for publication.

01/27/2024

Best wishes, Reviewer

Author Response

Dear reviewers:

Thank you very much for your acknowledgement and support of my manuscript. I am very pleased to hear that you consider my study to be of significant value and agree with the aims and methods of my research. We have further checked and revised the article in details and tried our best to improve the quality of my study.

Thank you again for your support and help. I look forward to receiving your guidance and encouragement again in my future academic path.

Best wishes!

With kind regards.

Ziwei Li